# $\mathbb{VITA}$: Video Instance Segmentation via Object Token Association

**Miran Heo**[*]
Yonsei University

**Sukjun Hwang**[*]
Yonsei University

**Seoung Wug Oh**
Adobe Research

**Joon-Young Lee**
Adobe Research

**Seon Joo Kim**
Yonsei University

{miran, sj.hwang, seonjookim}@yonsei.ac.kr          {seoh, jolee}@adobe.com

## Abstract

We introduce a novel paradigm for offline Video Instance Segmentation (VIS), based on the hypothesis that explicit object-oriented information can be a strong clue for understanding the context of the entire sequence. To this end, we propose $\mathbb{VITA}$, a simple structure built on top of an off-the-shelf Transformer-based image instance segmentation model. Specifically, we use an image object detector as a means of distilling object-specific contexts into object tokens. VITA accomplishes video-level understanding by associating frame-level object tokens without using spatio-temporal backbone features. By effectively building relationships between objects using the condensed information, VITA achieves the state-of-the-art on VIS benchmarks with a ResNet-50 backbone: 49.8 AP, 45.7 AP on YouTube-VIS 2019 & 2021, and 19.6 AP on OVIS. Moreover, thanks to its object token-based structure that is disjoint from the backbone features, VITA shows several practical advantages that previous offline VIS methods have not explored - handling long and high-resolution videos with a common GPU, and freezing a frame-level detector trained on image domain. Code is available at https://github.com/sukjunhwang/VITA.

## 1   Introduction

The goal of Video Instance Segmentation (VIS) is to predict both mask trajectories and categories of each object belonging to a set of predefined categories. Numerous studies have attained the goal in a variety of ways, but a notable innovation in terms of accuracy has been achieved by Transformer-based [27] architectures. Extending DETR [5] to the video domain, VisTR [28] made the first attempt to design an end-to-end model that jointly predicts object trajectories with their corresponding segmentation masks. By adopting this paradigm, subsequent studies [15, 30, 6, 34] also tackle the problem in a complete-offline manner: *video-in and video-out*.

The key message from the follow-up approaches [15, 30, 6, 34] is to effectively design core interactions between frames. In parallel with recent studies [11, 25, 37, 7, 22] that improve the accuracy in various tasks by localizing the attention scope of Transformer layers, the subsequent VIS methods suggest bounding the attention scope in the encoder [15, 34] or the decoder [30]. Specifically, they decompose the global attention by iteratively mixing two phases: intra-frame attention and inter-frame communication. Interestingly, the temporal interactions between frames are commonly achieved with only a small number of tokens, *e.g.,* memory tokens [15, 34], messenger tokens [34], and instance queries [30]. As a result, the question arises: "what information is important to understand a video?"

---

[*]Both authors contributed equally to this work.

36th Conference on Neural Information Processing Systems (NeurIPS 2022).

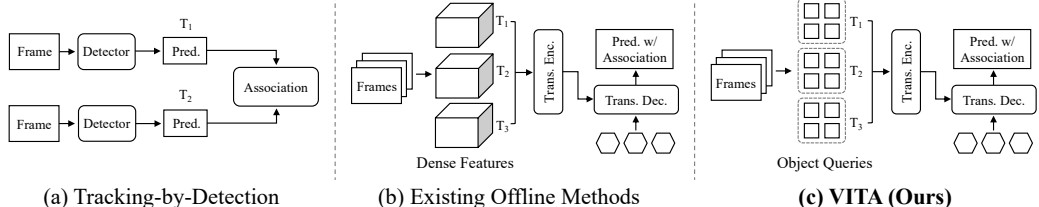

Figure 1: (a) Early-stage VIS methods divide the problem into two components, detection and association. (b) To alleviate the context-limited structure, complete-offline methods jointly track and segment instances in an end-to-end manner by employing dense spatio-temporal features. (c) On the other hand, our VITA is a new paradigm that directly leverages object queries for offline VIS.

In this paper, we introduce **V**ideo **I**nstance Segmentation via Object **T**oken **A**ssociation (**VITA**), a new offline VIS paradigm which suggests that a video can be effectively understood from a collection of object-centric tokens. Existing offline methods [28, 15, 30, 6, 34] (Fig. 1 (b)) localize objects in multiple frames by iteratively referring to dense spatio-temporal backbone features. However, such methods show difficulties in handling long sequences as the myriad of dense reference features hinders the Transformer layers from retrieving relevant information. With the motivation to devise an effective method for the long-range understanding, we obtain clues from the traditional tracking-by-detection paradigm (Fig. 1 (a)) and make two hypotheses: 1) an image object detector can fully embody the context of an object into a feature vector (or a token); and 2) a video can be represented by the relationship between the objects.

In this regard, VITA aims to parse an input video from the collection of object tokens without the necessity of referencing dense spatio-temporal backbone (Fig. 1 (c)). Given the compactness of the token representation, VITA can collect the object tokens over the whole video and directly analyzes the collection using Transformer layers. This unique design enables the complete-offline inference (i.e., video-in and video-out) even for extremely long videos. This also facilitates building relationships between every detected object and successfully achieves global video understanding. As a result, VITA achieves state-of-the-art performance on various VIS benchmarks.

We evaluate VITA on three popular VIS benchmarks, YouTube-VIS 2019 & 2021 [32] and OVIS [24]. With ResNet-50 [14] backbone, VITA achieves the new state-of-the-arts of 49.8 AP & 45.7 AP on YouTube-VIS 2019 & 2021, and 19.6 AP on OVIS. Above all, VITA outperforms the previous best approaches by 5.1 AP for YouTube-VIS 2021, which contains more complicated and long sequences than YouTube-VIS 2019. VITA is the first offline method that presents the results on OVIS benchmark that consists of long videos (the longest video has 292 frames) using a single 12GB GPU.

In addition to the performance, the design of VITA have several practical advantages over the previous offline VIS methods. It can handle long and high-resolution videos so it does not require heuristics for associating clip-level results. VITA can process 1392 frames at once regardless of video resolution using a single 12GB GPU which is 11 times longer than IFC [15]. Moreover, VITA can be trained on top of a parameter-frozen image object detector without sacrificing the performance much. This property is especially useful for the applications that cannot afford to store separated image and video instance segmentation models. VITA takes only 6% additional parameters to extend the Swin-L detector.

## 2 Related Works

**Online VIS** approaches first predict individual tracklets within a local range window consisting of a single or a few frames. After obtaining results from adjacent windows, they associate individual tracklets of same identities by a hand-crafted or a learnable matching algorithm. MaskTrack R-CNN [32] sets the groundwork for VIS research by proposing a simple tracking branch added on a two-stage image instance segmentation model [13]. The methods [4, 33, 21] that follow the tracking-by-detection paradigm (Fig. 1 (a)) measure the similarities between per-frame predictions, then employ an association algorithm.

To deploy temporal context from multiple frames, per-clip methods [1, 2] design an architecture of predicting tracklets within a local window and stitching the tracklets sequentially in a near-

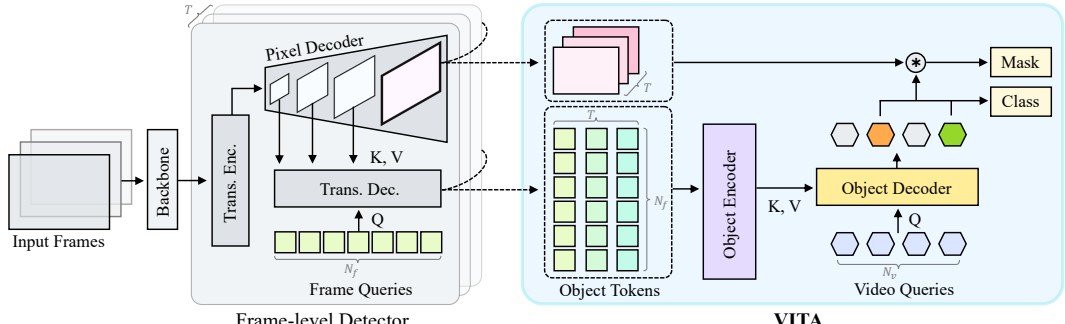

Figure 2: VITA takes only mask features and frame queries that are independently decoded by the frame-level detector for entire video sequence. By directly constructing temporal interactions between frame queries that encapsulate rich object-aware knowledge in spatial scenes, VITA yields mask trajectories with corresponding categories in an end-to-end manner.

online manner. Propagation-based methods [10, 17, 12] devise a paradigm that conjugates rich previous information stored in memories to facilitate online applications. EfficientVIS [29] introduces correspondence learning between adjacent tracklet features and successfully runs in a cascaded manner which eliminates the hand-crafted tracklet association.

**Offline VIS** architectures are proposed with the motivation of predicting mask trajectories through a whole video sequence at once. VisTR [28] successfully extends DETR [5] to the VIS domain, introducing a new paradigm of jointly tracking and segmenting instances. However, its dense self-attention over the spatio-temporal inputs leads to explosive computations and memories. With the motivation of relaxing the heavy computation of VisTR, IFC [15] adopts memory tokens to the Transformer encoder and decodes clip-level object queries. By setting the frame-level encoder to be independent and adopting the decoder of IFC, Mask2Former-VIS [6] records considerable performance on benchmarks by taking the advantage of its mask-oriented representation [7]. TeViT [34] proposes a new backbone that efficiently exchanges temporal information internally based on Vision Transformers [9] instead of the frame-wise CNN backbone. SeqFormer [30] decomposes the decoder to be frame-independent, while building communication between different frames using instance queries that are used for frame-wise detection. All these studies achieve promising performance by referring to dense backbone features (Fig. 1 (b)). On the other hand, our VITA suggests a new offline VIS paradigm that directly interprets a video from the collection of object tokens (Fig. 1 (c)).

**Global trackers** that aim to associate frame-level predictions across an entire sequence as a whole are studied in the Multiple Object Tracking (MOT) community. Conventional approaches formulate the problem as a graph optimization – interpreting each detection as a node and considering the edges as possible connections between the nodes [35, 26, 3, 8]. Different from existing methods, GTR [36] introduces a Transformer-based architecture that receives queries, then explicitly searches for the predictions with the same identities. Similarly, a recent method [16] proposes a set classifier that classifies the category of each tracklet by globally aggregating information from multiple frames.

## 3 Method

In this section, we first give a brief overview of Mask2Former [7], a frame-level detector for VITA. Then, we introduce the architecture of our proposed VITA, which is built on top of Mask2Former. Finally, we describe how VITA handles extremely long videos in a complete-offline manner.

### 3.1 Frame-level Detector

In this paper, we adopt Mask2Former [7] for the frame-level detector which directly localizes instances using masks without the necessity of bounding boxes. Following the set prediction mechanism of DETR [5], the frame-level detector parse an input image $H \times W$ using $N_f$ object queries, which we call *frame queries* ($f \in \mathbb{R}^{C \times N_f}$) throughout this paper. Having the spatially encoded features to be decoded by the frame queries through a Transformer decoder, each object in the image gets represented as a $C$-dimensional vector. Then, the frame queries are used for both classifying and

segmenting their matched objects where the predictions are also used for auxiliary supervision for VITA. Specifically, the frame-level detector generates two features for the frame-level predictions: 1) dynamic $1 \times 1$ convolutional weight from the frame queries; 2) per-pixel embeddings $\mathcal{M} \in \mathbb{R}^{C \times \frac{H}{S} \times \frac{W}{S}}$ from the pixel decoder, where $S$ is the stride of the feature map. Finally, the detector segments objects by applying a simple dot product between the two embeddings.

## 3.2 VITA

We now propose the novel end-to-end video instance segmentation method VITA, which can be largely divided into three phases (Fig. 2). First, VITA operates on top of the frame-level detector [7] in a complete frame-independent manner; no inter-computation between frames is involved. Then, the frame queries that hold object-centric information are collected throughout the whole video and they embed video-level information by building communications between different frames using Object Encoder. Finally, Object Decoder aggregates information from the frame queries to video queries, which are eventually used for predicting categories and masks of objects in videos at once.

**Input of VITA.** Given an input video of $T$ frames, the frame-level detector executes frame-by-frame as previously explained. Among a number of intermediate embeddings that are generated by the detector, the only features that are used by VITA are 1) frame queries $\{f^t\}_{t=1}^T \in \mathbb{R}^{C \times T \times N_f}$ which hold object-centric information; and 2) per-pixel embeddings $\{\mathcal{M}^t\}_{t=1}^T \in \mathbb{R}^{C \times T \times \frac{H}{S} \times \frac{W}{S}}$ from the pixel decoder.

**Object Encoder.** After the frame-wise detector distills the object-wise context into the frame queries, Object Encoder aims to build temporal communication by employing self-attention along the temporal axis. First, Object Encoder gathers frame queries from all frames and converts them to object tokens through a linear layer. However, a naive self-attention over the whole $TN_f$ object tokens is not applicable when processing long videos due to the quadratic computational overhead of Transformers. Inspired by Swin Transformer [22], we adopt window-based self-attention layers that shift along the temporal dimension. As illustrated in Fig. 3, Object Encoder

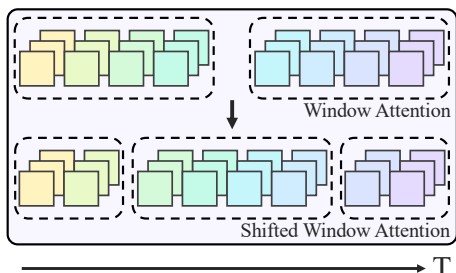

Figure 3: Illustration of an Object Encoder layer. Blocks with dashed line are local windows, and □ indicates an object token.

initially partitions object tokens $\{f^t\}_{t=1}^T$ to the temporal axis with local windows of size $W$ without an overlap. By alternatively shifting the windows, object tokens from different frames can exchange object-wise information which allows VITA to both effectively and efficiently handle long sequences.

**Object Decoder and Output heads.** Two limitations of previous offline VIS methods [28, 15, 6] are the ineffectiveness in handling dynamic scenes and the inability of processing long videos. For example, such methods obtain high accuracy when dealing with static and short videos (YouTube-VIS 2019 [32]), but struggle to track objects or executes end-to-end on benchmarks with dynamic and long videos (YouTube-VIS 2021 [32] and OVIS [24]). Both limitations are mainly caused by the decoder, which parses object contexts directly from dense spatio-temporal features. As recent studies [11, 25, 37] suggest, typical Transformer decoders show difficulties in retrieving relevant information from global context. In the video domain, the number of backbone features being referred to proportionally increases with the number of frames. Therefore, when handling extremely long videos, the countless reference tokens result in both imprecise information retrieval and intractable peak memories.

For the solution to the problem, we suggest Object Decoder which extracts information from the object tokens, not the spatio-temporal backbone features. Implicitly embedding the context of objects, object tokens can provide sufficient instance-specific information without the interference of dense backbone features. Specifically, we employ $N_v$ trainable video queries $v \in \mathbb{R}^{C \times N_v}$ to decode object-wise information from all object tokens $\{f^t\}_{t=1}^T$ that are collected from all $T$ frames. Receiving much condensed input over naively taking dense spatio-temporal features, Object Decoder effectively captures video contexts and aggregates relevant information into the video queries. As a result, Object

Decoder shows fast convergence speed while achieving high accuracy. Furthermore, the compact input greatly saves memories, thus facilitates processing long and high-resolution videos.

From the decoded video queries $v$, VITA returns final predictions $z = \{(p_i, m_i)\}_{i=1}^{N_v}$ using two output heads similar to IFC [15]; the class head and the mask head. The class head is a single linear classifier, which directly predicts class probabilities $p \in \mathbb{R}^{N_v \times (K+1)}$ of each video query, where $K + 1$ is the number of categories including an auxiliary label "no object" ($\varnothing$). The mask head dynamically generates mask embeddings $w_v \in \mathbb{R}^{C \times N_v}$ per a video query, which corresponds to the tracklet of an instance over all frames. Finally, the predicted mask logits $m \in \mathbb{R}^{N_v \times T \times H \times W}$ can be obtained from a matrix multiplication between $w_v$ and $\{\mathcal{M}^t\}_{t=1}^{T}$.

### 3.3 Clip-wise losses

**Instance matching.** We search for optimal pair indices between the predictions from VITA and $G_v$ ground-truth to remove post-processing heuristics such as NMS. First, we calculate costs from all possible pairs using the cost function of Mask2Former [7] with a simple extension of mask-related costs to the temporal axis [15]. Then, from $N_v \times G_v$ costs of pairs, we follow DETR [5] and use Hungarian algorithm [18] for the optimal matching as shown in Fig. 4 (b).

**Similarity loss.** Inspired by the initial VIS approach (MaskTrack R-CNN [32]) where the similarity loss is adopted to track instances at different frames, we train video queries and frame queries to be clustered in the latent space by their identities. As shown in Fig. 4 (a), our adopted frame-level detector [7] also searches for paired indices be-

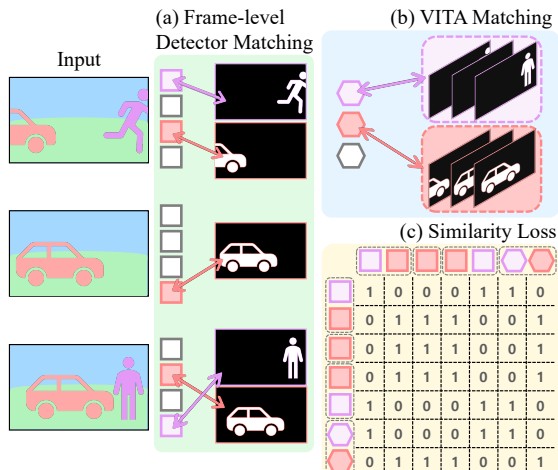

Figure 4: Similarity loss. ○ and □ indicate video query and frame query, respectively. Same color represents same GT instance ID.

tween $N_f$ frame-wise predictions and $G_f^t$ ground-truth objects at each $t^{\text{th}}$ frame. The frame queries and the video queries that are matched to ground-truths get collected and we embed the collection through a linear layer. Then, we measure the similarity of all possible pairs using a simple matrix multiplication. Finally, as shown in Fig. 4 (c), binary cross entropy is used to compute $\mathcal{L}_{sim}$ between the predicted similarities and the ground-truth where annotated to 1 for pairs of equal identities and 0 for vice-versa.

**Total loss.** We attach the proposed module VITA on top of the frame-level detector, and all components of the model get trained end-to-end. Note that not only video-level outputs from VITA are used for the loss computation, but also per-frame outputs from the frame-level detector get involved. Specifically, we use $\mathcal{L}_f$ from [7] to calculate loss from the per-frame outputs to frame-wise ground-truth. Extending the loss function of [7] to the temporal axis as similar to [15], we use outputs from VITA $z$ to calculate the video-level loss $\mathcal{L}_v$. Finally, we integrate all losses together as follows: $\mathcal{L}_{total} = \lambda_v \mathcal{L}_v + \lambda_f \mathcal{L}_f + \lambda_{sim} \mathcal{L}_{sim}$.

## 4 Experiments

### 4.1 Datasets

**YouTube-VIS 2019.** YouTube-VIS 2019 [32] is the first dataset proposed for VIS and contains 40 semantic categories. Mostly originated from Video Object Segmentation (VOS) datasets, the VIS benchmark has a small number of unique instances (average 1.7 per video for the `train` set) and the categories of instances appearing in the same video are different in general. Also, the average length of videos in the `valid` set is short (27.4 frames), which enables existing complete-offline approaches to load a whole video and infer the benchmark at once.

Table 1: Comparisons on YouTube-VIS 2019.

| Method | | Backbone | AP | $AP_{50}$ | $AP_{75}$ | $AR_1$ | $AR_{10}$ |
|---|---|---|---|---|---|---|---|
| (Near) Online | MaskTrack R-CNN [32] | ResNet-50 | 30.3 | 51.1 | 32.6 | 31.0 | 35.5 |
| | MaskTrack R-CNN [32] | ResNet-101 | 31.8 | 53.0 | 33.6 | 33.2 | 37.6 |
| | CrossVIS [33] | ResNet-50 | 36.3 | 56.8 | 38.9 | 35.6 | 40.7 |
| | CrossVIS [33] | ResNet-101 | 36.6 | 57.3 | 39.7 | 36.0 | 42.0 |
| | PCAN [17] | ResNet-50 | 36.1 | 54.9 | 39.4 | 36.3 | 41.6 |
| | PCAN [17] | ResNet-101 | 37.6 | 57.2 | 41.3 | 37.2 | 43.9 |
| | EfficientVIS [29] | ResNet-50 | 37.9 | 59.7 | 43.0 | 40.3 | 46.6 |
| | EfficientVIS [29] | ResNet-101 | 39.8 | 61.8 | 44.7 | 42.1 | 49.8 |
| | VISOLO [12] | ResNet-50 | 38.6 | 56.3 | 43.7 | 35.7 | 42.5 |
| Offline | VisTR [28] | ResNet-50 | 35.6 | 56.8 | 37.0 | 35.2 | 40.2 |
| | VisTR [28] | ResNet-101 | 38.6 | 61.3 | 42.3 | 37.6 | 44.2 |
| | IFC [15] | ResNet-50 | 41.2 | 65.1 | 44.6 | 42.3 | 49.6 |
| | IFC [15] | ResNet-101 | 42.6 | 66.6 | 46.3 | 43.5 | 51.4 |
| | TeViT [34] | MsgShifT | 46.6 | 71.3 | 51.6 | 44.9 | 54.3 |
| | SeqFormer [30] | ResNet-50 | 47.4 | 69.8 | 51.8 | 45.5 | 54.8 |
| | SeqFormer [30] | ResNet-101 | 49.0 | 71.1 | 55.7 | 46.8 | 56.9 |
| | SeqFormer [30] | Swin-L | 59.3 | 82.1 | 66.4 | 51.7 | 64.4 |
| | Mask2Former-VIS [6] | ResNet-50 | 46.4 | 68.0 | 50.0 | - | - |
| | Mask2Former-VIS [6] | ResNet-101 | 49.2 | 72.8 | 54.2 | - | - |
| | Mask2Former-VIS [6] | Swin-L | 60.4 | 84.4 | 67.0 | - | - |
| VITA (Ours) | | ResNet-50 | 49.8 | 72.6 | 54.5 | 49.4 | 61.0 |
| | | ResNet-101 | 51.9 | 75.4 | 57.0 | 49.6 | 59.1 |
| | | Swin-L | 63.0 | 86.9 | 67.9 | 56.3 | 68.1 |

**YouTube-VIS 2021.**  In order to address more difficult scenarios, additional videos are included in YouTube-VIS2021 (794 videos for training and 129 videos for validation). In particular, a greater number of objects with confusing trajectories has been added (average 3.4 per video for the additional videos in the `train` set). However, the average length of the additional validation videos is still 39.7 frames, which is not significantly increased compared to YouTube-VIS 2019.

**OVIS.**  Under the same definition as YouTube-VIS, OVIS [24] specifically aims to tackle objects with heavy occlusions that are belonging to 25 semantic categories. In addition to the heavily occluded situation, OVIS has three challenging characteristics that are distinct from the YouTube-VIS datasets. First, although it has fewer categories than YouTube-VIS, much more instances appear in a single video (average 5.9 per video for the `train` set). Second, the instances with the same categories in the same video have almost similar appearances, thus approaches that rely heavily on visual cues often struggle to predict accurate trajectories. Finally, the average length of videos for the `valid` set is 62.7 frames (the longest video has 292 frames) which is much longer than that of YouTube-VIS. Therefore, not only do previous approaches show relatively low accuracy, but all existing complete-offline VIS methods are not feasible to infer OVIS without hand-crafted association algorithms.

### 4.2 Implementation Details

Our method is implemented on top of `detectron2` [31]. All hyper-parameters regarding the frame-level detector are equal to the defaults of Mask2Former [7]. The total loss $\mathcal{L}_{total}$ is balanced with $\lambda_v$, $\lambda_f$, and $\lambda_{sim}$ where 1.0, 1.0, and 0.5, respectively. By default, Object Encoder is composed of three layers with the window size $W = 6$, and Object Decoder employs six layers with $N_v = 100$ video queries. Having VITA built on top of Mask2Former, we first train our model on the COCO [20] dataset following Mask2Former. Then, we train our method on the VIS datasets [32, 24] simultaneously with pseudo videos generated from images [20] following the details of SeqFormer [30]. During inference, each frame is resized to a shorter edge size of 360 and 448 pixels when using ResNet [14] and Swin [22] backbones, respectively. Note that all reported scores in main results and ablation

Table 2: Comparisons with ResNet-50 backbone on YouTube-VIS 2021 and OVIS. † indicates using MsgShifT backbone. ‡ indicates using Swin-L [22] backbone.

| Method | YouTube-VIS 2021 | | | | | OVIS | | | | |
|---|---|---|---|---|---|---|---|---|---|---|
| | AP | $AP_{50}$ | $AP_{75}$ | $AR_1$ | $AR_{10}$ | AP | $AP_{50}$ | $AP_{75}$ | $AR_1$ | $AR_{10}$ |
| MaskTrack R-CNN [32] | 28.6 | 48.9 | 29.6 | 26.5 | 33.8 | 10.8 | 25.3 | 8.5 | 7.9 | 14.9 |
| CMaskTrack R-CNN [23] | - | - | - | - | - | 15.4 | 33.9 | 13.1 | 9.3 | 20.0 |
| STMask [19] | 31.1 | 50.4 | 33.5 | 26.9 | 35.6 | 15.4 | 33.8 | 12.5 | 8.9 | 21.3 |
| CrossVIS [33] | 34.2 | 54.4 | 37.9 | 30.4 | 38.2 | 14.9 | 32.7 | 12.1 | 10.3 | 19.8 |
| IFC [15] | 35.2 | 55.9 | 37.7 | 32.6 | 42.9 | - | - | - | - | - |
| VISOLO [12] | 36.9 | 54.7 | 40.2 | 30.6 | 40.9 | 15.3 | 31.0 | 13.8 | 11.1 | 21.7 |
| TeViT† [34] | 37.9 | 61.2 | 42.1 | 35.1 | 44.6 | 17.4 | 34.9 | 15.0 | 11.2 | 21.8 |
| SeqFormer [30] | 40.5 | 62.4 | 43.7 | 36.1 | 48.1 | - | - | - | - | - |
| Mask2Former-VIS [6] | 40.6 | 60.9 | 41.8 | - | - | - | - | - | - | - |
| **VITA (Ours)** | **45.7** | **67.4** | **49.5** | **40.9** | **53.6** | **19.6** | **41.2** | **17.4** | **11.7** | **26.0** |
| SeqFormer‡ [30] | 51.8 | 74.6 | 58.2 | 42.8 | 58.1 | - | - | - | - | - |
| Mask2Former-VIS‡ [6] | 52.6 | 76.4 | 57.2 | - | - | - | - | - | - | - |
| **VITA (Ours)‡** | **57.5** | **80.6** | **61.0** | **47.7** | **62.6** | **27.7** | **51.9** | **24.9** | **14.9** | **33.0** |

studies are the mean of five runs, and we use the standard ResNet-50 [14] for the backbone unless specified.

## 4.3 Main Results

Using the popular VIS benchmarks – YouTube-VIS 2019 & 2021 [32] and OVIS [24] – we compare VITA with state-of-the-art approaches following the standard evaluation metric [32].

**YouTube-VIS 2019.** Tab. 1 shows the comparison on YouTube-VIS 2019 dataset with backbones of both CNN-based (ResNet-50 and 101 [14]) and Transformer-based (Swin-L [22]). Offline methods can take two advantages over (near) online approaches: 1) they have a greater receptive field to the temporal axis, and 2) they can avoid error propagation derived from hand-crafted association algorithms. As a result, the tendency of offline methods with higher accuracy is clearly shown in the table. Among the competitive offline models, our VITA sets a new state-of-the-art of 49.8 AP and 51.7 AP using CNN backbones, ResNet-50 and ResNet-101 respectively. In addition, with Swin-L backbone, VITA achieves 63.0 AP outperforming all existing VIS methods.

**YouTube-VIS 2021.** We compare VITA with state-of-the-art methods on YouTube-VIS 2021 benchmark in Tab. 2. Note that the longest video in the `valid` set has 84 frames, thus previous offline methods [15, 30, 6] can infer videos at once with GPUs with large memories. Above all, VITA achieves the highest accuracy, 45.7 AP, which outperforms the previous state-of-the-art approach [6] with a huge margin of 5.1 AP. Considering the accuracy gap on YouTube-VIS 2019, the results demonstrate that VITA can effectively handle tricky scenarios, *e.g.,* numerous unique instances with confusing trajectories. We hypothesize that the object-oriented design of VITA is more effective than typical dense Transformer decoders in addressing such challenging scenes.

**OVIS.** In Tab. 2, we demonstrate the competitiveness of VITA on the challenging OVIS benchmark. Due to the considerable lengths of videos – the longest video has 292 frames – existing offline approaches [28, 15, 34, 30, 6] cannot process OVIS benchmark in their original design: video-in and video-out. To the best of our knowledge, VITA is the first complete-offline approach to evaluate on OVIS `valid` set. Thanks to its object token-based structure which is disjoint from backbone features, VITA can process the benchmark *without* any hand-crafted association algorithm. Moreover, VITA sets a new state-of-the-art performance of 19.6 AP, demonstrating the potential of the complete-offline pipeline in long and complicated scenes.

Table 3: Impact of local windows of varying sizes in Object Encoder.

| $W$ | AP | $AP_{50}$ | $AP_{75}$ | $AR_1$ | $AR_{10}$ |
|---|---|---|---|---|---|
| 3 | 49.4 | 72.2 | 54.4 | 48.6 | 60.9 |
| 6 | 49.8 | 72.6 | 54.5 | 49.4 | 61.0 |
| 12 | 50.0 | 73.0 | 54.7 | 49.0 | 60.8 |
| All | 50.1 | 72.4 | 54.7 | 49.0 | 60.6 |

Table 4: Maximum number of frames that can be processed at once using a single Titan XP.

| Method | Max Frames | |
|---|---|---|
| | $360 \times 640$ | $720 \times 1280$ |
| VisTR [28] | 46 | 12 |
| IFC [15] | 123 | 38 |
| Mask2Former-VIS [6] | 81 | 20 |
| VITA (Ours) $W = 3$ | 2677 | |
| $W = 6$ | 1392 | |
| $W = 12$ | 741 | |

Table 5: Use of different heuristic association algorithms on OVIS `valid` set.

| Length | Algorithm | AP | $AP_{50}$ | $AP_{75}$ |
|---|---|---|---|---|
| 36 | Greedy | 18.8 | 39.4 | 17.1 |
| | Hungarian | 18.4 | 38.9 | 16.3 |
| 48 | Greedy | 18.8 | 39.0 | 17.1 |
| | Hungarian | 19.1 | 39.1 | 17.4 |
| All | None | 19.6 | 41.2 | 17.4 |

Table 6: Pruning tokens by different ratios $r$.

| $r$ | AP | $AP_{50}$ | $AP_{75}$ | $AR_1$ | $AR_{10}$ |
|---|---|---|---|---|---|
| 1.0 | 49.8 | 72.6 | 54.5 | 49.4 | 61.0 |
| 0.75 | 49.7 | 72.5 | 54.4 | 48.7 | 61.0 |
| 0.5 | 48.9 | 72.1 | 52.0 | 48.3 | 60.9 |
| 0.25 | 48.1 | 71.6 | 51.6 | 47.4 | 59.8 |

### 4.4 Ablation Studies

We provide a series of ablation studies using a ResNet-50 [14] backbone. All experiments are conducted on YouTube-VIS 2019 [32] `valid` set except for Tab. 5 with OVIS [24] `valid` set.

**Attention window size.** Tab. 3 shows the performance of VITA with varying sizes of shifted attention window $W$ in Object Encoder during inference. The larger the window, the greater the receptive field for the temporal axis in Object Encoder. The results suggest that larger window sizes utilize information from multiple frames, which helps Object Encoder understand the context of objects in videos. We set $W = 6$ considering a trade-off between performance and inference scalability.

**Maximum number of frames.** In Tab. 4, we calculate the maximum number of frames that VITA can handle with respect to the various window sizes $W$, and compare it with existing complete-offline VIS methods. To take into account the general environment, all results are computed using a single 12GB Titan XP GPU. As shown in results, existing methods have limitations in processing long videos in a *video-in and video-out* manner. Clearly, the bottleneck of VisTR [28] is the encoder, where the full spatio-temporal self-attention leads to a tremendous memory usage. IFC [15] alleviates the computation of VisTR [28], achieving a higher number of input frames. However, IFC makes use of a typical Transformer decoder that visits all dense spatio-temporal features. Therefore, IFC cannot infer the OVIS [24] benchmark at once which contains a video of 292 frames. The problem gets aggravated in Mask2Former-VIS [6] as the scope of the decoder is extended to multiple feature levels [7]. On the other hand, VITA presents considerable frame numbers that can be inferred completely offline. Furthermore, VITA is independent from input frame resolutions as each frame gets summarized into compact object tokens. With input resolution of $360 \times 640$ and $W = 6$, the maximum length of sequence that VITA is able to process in complete-offline is about *11× longer* than IFC [15].

**Heuristic clip association.** Tab. 5 shows the results on OVIS `valid` set of splitting a video into shorter clips and associating clip-wise predictions through heuristic matching. The length of the clip is set to be less than the average length of videos of OVIS `valid` set (62.7). Then, we associate outputs from different clips using mask IoU score as the matching cost. We test with two matching algorithms: Greedy and Hungarian. As shown in Tab. 5, VITA demonstrates the best performance on the complete-offline inference that use all the video frames at once.

**Pruning Tokens.** In Tab. 6, we investigate the effects of removing redundant frame queries. From a collection of frame queries, VITA understands the overall context of the given clip. As only a small portion of the collection is matched to foreground objects, the number of total input frame queries

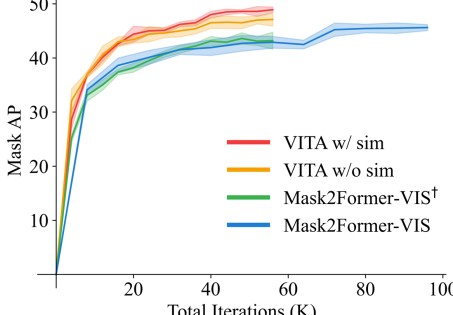

Figure 5: Train speed comparison with Mask2Former-VIS [6]. † indicates the same training setup with VITA.

Table 7: Results on YouTube-VIS 2019 with freezing detector pretrained on COCO.

| Backbone | Freeze | AP | $AP_{50}$ | $AP_{75}$ | $AR_1$ | $AR_{10}$ |
|---|---|---|---|---|---|---|
| ResNet-50 | | 49.8 | 72.6 | 54.5 | 49.4 | 61.0 |
| | ✓ | 40.9 | 61.9 | 44.6 | 43.1 | 53.1 |
| ResNet-101 | | 51.9 | 75.4 | 57.0 | 49.6 | 59.1 |
| | ✓ | 43.2 | 64.4 | 48.7 | 46.1 | 55.9 |
| Swin-L | | 63.0 | 86.9 | 67.9 | 56.3 | 68.1 |
| | ✓ | 53.4 | 75.9 | 58.7 | 51.9 | 64.3 |

can be reduced. First, for each frame, we sort frame queries in ascending order by the "no object" ($\varnothing$) probability. Then, we keep only top $rN_f$ queries from the sorted list where $r$ is the ratio, and discard the rest. The accuracy with respect to the ratio $r$ is as shown in Tab. 6.

By setting the ratio $r = 0.75$, the accuracy of VITA shows only a marginal degradation in the accuracy ($-0.1$ AP). This signifies that VITA focuses more on the foreground contexts that are embedded in the frame queries. Meanwhile, as the quadratic computation in Clip Encoder can be alleviated, VITA can process a much greater number of frames; using the ratio $r = 0.75$, the maximum frame number increases from 1392 (Tab. 4) to 2635.

**Convergence speed and Similarity loss.** Fig. 5 validates our claim of the faster convergence speed and the effectiveness of the proposed Similarity loss. For a fair comparison, we report average scores and standard deviations of five runs, each trained without pseudo videos, same as Mask2Former-VIS [6]. Thanks to its object-centric design, VITA shows faster convergence than Mask2Former-VIS. Furthermore, the use of Similarity loss leads to an additional accuracy gain of 1.8 AP. The results demonstrate that the loss mitigates the discrepancies between the embeddings of equal identities, leading to better performance.

**Frozen frame-level detector.** In Tab. 7, we demonstrate the performance of VITA where the frame-level detector is completely frozen. Specifically, while VITA gets trained on YouTube-VIS 2019, the frame-level detector [7] does not get updated from pretrained weights on COCO [20]. Note that among 40 categories in YouTube-VIS 2019 dataset, only 20 categories overlap with the categories of COCO. Interestingly, though the frame-level detector remains completely frozen, VITA achieves compelling results with various backbones. As shown in Tab. 1 and Tab. 7, VITA presents a huge practicality as it surpasses all online approaches on top of the ResNet-50 backbone. This strategy can be beneficial in various scenarios: 1) when the accuracy of image instance segmentation should be kept while extending the network to the video domain, and 2) when having limited time and GPUs to train models. The strategy can be especially useful in mobile applications that have scarce storage for keeping two separate network parameters for image instance segmentation and video instance segmentation. With additional 6% parameters, VITA successfully extends the frozen Swin-L based frame-level detector to the video domain and it achieves great accuracy.

We also provide a brief discussion of our understanding for the large gap in AP. Compared to COCO, we observe that YouTube-VIS dataset is annotated with only a few salient objects as foregrounds. Having weights of the frame-level detector frozen to COCO, the detector cannot adapt to the YouTube-VIS domain and it embeds and interprets more objects in scenes as foregrounds. Therefore, VITA outputs more predictions as a foreground category even if such predictions are not labeled as ground-truths in YouTube-VIS. As a result, it leads to a lower average precision as it comes out with more false positive predictions. On the contrary, the more false positive predictions only slightly affect AR.

**Qualitative Results.** We provide some visualizations of the predictions from VITA and frame-level detector in Fig. 6. The qualitative results show that VITA leads to better video instance segmentation qualities compared to the frame-level detector. Specifically, the frame-level detector mistakenly interprets in to recognize either category or mask of instances that have been largely occluded, while our method successfully recovers it by leveraging the temporal information.

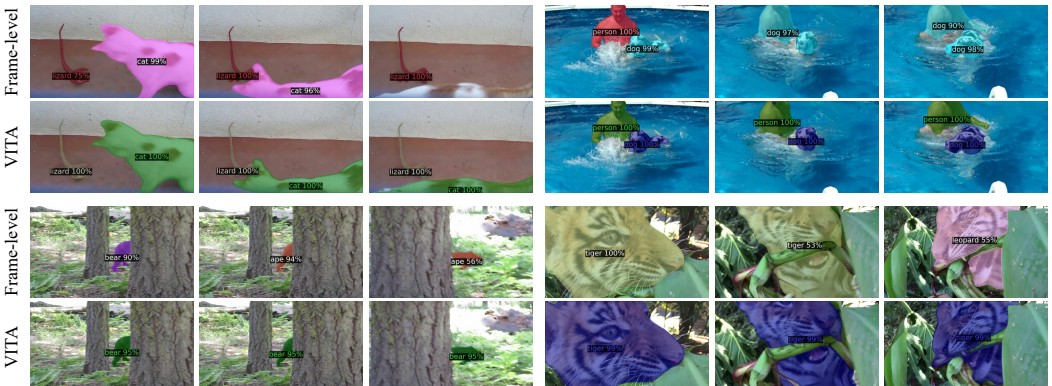

Figure 6: Visualization of predictions from the frame-level detector and VITA. Instances with the same identity are displayed in the same color.

# 5   Limitations

VITA has achieved high performance in the complete-offline paradigm while dramatically improving the number of input frames that can be processed at once. However, there are two major limitations for the ultimate long video understanding. First, the current architecture still has limitations in processing an infinite number of frames. In addition, since object tokens do not explicitly utilize temporal information, they may have difficulties in identifying complex behaviors that span over very long sequences. We believe that devising explicit designs to address these issues will be a promising future direction.

# 6   Conclusion

In this paper, we proposed VITA for offline Video Instance Segmentation. VITA is a simple model built on top of the off-the-shelf image instance segmentation model [7]. Unlike existing offline methods, VITA directly leverages object queries decoded by independent frame-level detectors. We demonstrated that deploying object-oriented information is not only effective in improving performance, but also has robust practicality for processing long and high-resolution videos - setting state-of-the-art on popular VIS benchmarks, *e.g.*, YouTubeVIS-2019 & 2021 and OVIS. Moreover, since VITA is designed to absorb spatial knowledge purely from image object detector, it shows fast convergence and demonstrates competitive performance even if trained on frozen detectors. We hope that our method extends the scope of offline VIS research beyond benchmarks to real-world applications.

## Acknowledgements

This work has partly supported by the National Research Foundation of Korea (NRF) grant funded by the Korea government (MSIT) (NRF2022R1A2C2004509) and by Institute of Information communications Technology Planning Evaluation (IITP) grant funded by the Korea government (MSIT) (No. 2022-0-00113, Developing a Sustainable Collaborative Multi-modal Lifelong Learning Framework), and Artificial Intelligence Graduate School Program under Grant 2020-0-01361.

## Broader Impact

VITA is designed for the VIS task and focuses on processing long and high-resolution videos in an end-to-end manner while achieving the state-of-the-art performance. We hope that VITA can have a positive impact on many industrial areas such as video editing applications. We would like to note that research on VIS must be aware of potential misuse that violates personal privacy.

**Licenses of COCO [20], YouTube-VIS [32], OVIS [24], and** `detectron2` **[31]:**   Attribution 4.0 International, CC BY 4.0, CC BY-NC-SA 4.0, and Apache-2.0, respectively.

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
