# OpenReview forum: "VITA: Video Instance Segmentation via Object Token Association"
_NeurIPS.cc/2022/Conference — NeurIPS 2022 Accept_

### Official Review · Reviewer_KReH · 2022-07-10

**Rating:** 7
**Confidence:** 5
**Soundness:** 3 good
**Presentation:** 3 good
**Contribution:** 3 good

**Summary:**

This paper introduces a novel paradigm for offline video instance segmentation tasks through explicitly modeling object-oriented queries. Specifically, this paper uses an image detector to distill object-specific contexts into object tokens, then accomplishes video-level understanding by associating frame-level object tokens with an extra object encoder. Experiments on YouTuve-VIS-2019, YouTuve-VIS-2021, and OVIS demonstrate the effectiveness of the proposed method.

**Questions:**

In general, I think this paper is good with clear writing, a well-explained approach, and good experimental results. The only concern is an unfair comparison with related works.

**Ethics Review Area:**

["I don’t know"]

**Limitations:**

There are no particular limitations in general. Please see the weakness above.

**Strengths And Weaknesses:**

Strengths:
+ Using object tokens to directly model spatio-temporal is novel and efficient.
+ This paper is well written and easy to follow. In addition, the authors also provide the code to reproduce.
+ Experiments on the VIS benchmark demonstrate the effectiveness of the proposed method.
+ Long-range video understanding is a crucial topic in video object detection, multi-object tracking, and video instance segmentation. This paper provides a new perspective for such areas.

Weakness:
+ Since this paper uses mask2former as image instance segmentation, thus it is not fair to compare with the previous SOTA method SeqFormer. I would like to see the results using image instance segmentation as the same as SeqFormer.
+ Using all image queries may be redundant. So, I suggest abandoning background queries and only using foreground queries for such video instance segmentation tasks. The number of different frames may not be equal. Thus, an alternative way is to use part of object queries to ensure each frame has an equal number of queries.
+ Why not provide YouTube-VIS-2022 experimental results?

---

> ### Author Response · Authors · 2022-08-02
> **Response to Reviewer KReH**
>
> We sincerely appreciate the reviewer for the thoughtful review and constructive feedback. We are pleased to see that the reviewer acknowledged our work “introduces novel and efficient ideas”, “is well written and easy to follow”, “demonstrates effectiveness on benchmarks”, and “provides a new perspective for long-range video understanding”. Our answers to the questions are as follows.
>
> ---
>
> **Q1. Unfair comparison with related works, especially SeqFormer.**
>
> We attached SeqFormer [28] on top of Mask2Former [7] detector (Mask2Former+Seqformer), and we provide a quantitative comparison between Mask2Former-VIS, Mask2Former+Seqformer, and VITA. The models are trained and evaluated on YouTube-VIS 2019 dataset without the use of pseudo videos.
>
> |Method                         |AP|AP$_{50}$|AP$_{75}$|AR$_{1}$|AR$_{10}$|
> |:---                           |:---:|:---:|:---:|:---:|:---:|
> |Mask2Former-VIS [6]            |46.4|68.0|50.0|46.4|56.4|
> |Mask2Former + SeqFormer [7, 28]|47.4|71.0|52.5|46.5|57.0|
> |VITA (Ours)                    |48.1|69.4|52.9|48.3|60.3|
>
> **Q2. Suggestion of removing redundant queries.**
>
> We greatly thank the reviewer for sharing the interesting idea. The pruning strategy is as follows:
>
> 1. The “background” probability ***p*** of all frame queries ***TN$_f$*** are measured.
> 2. ***p*** is sorted in ascending order.
> 3. For each frame, only top ***rN$_f$*** queries are sampled, where ***r*** **\in (0, 1]** is a ratio. (***p*** **=** ***p*** **[ :** ***rN$_f$*** **]**)
>
> The accuracy per ratio ***r*** is as shown in the following table:
>
> |***r***|AP     |AP$_{50}$|AP$_{75}$|AR$_{1}$|AR$_{10}$|
> |:---   |:---:  |:---:      |:---:      |:---:      |:---:|
> |1.0    |49.8   |72.6       |54.5       |49.4       |61.0|
> |0.75   |49.7   |72.5       |54.4       |48.7       |61.0|
> |0.5    |48.9   |72.1       |52.0       |48.3       |60.9|
> |0.25   |48.1   |71.6       |51.6       |47.4       |59.8|
>
> From the reduction of the number of the frame queries, VITA can process much more frames as the quadratic computation in Clip Encoder can be alleviated. By setting the ratio ***r*** **= 0.75**, the maximum frame number that VITA can process increases **from 1392 (Table. 4) to 2635**. Meanwhile, the accuracy degradation is marginal (**-0.1 AP**), which signifies that VITA focuses more on the foreground context. We believe exploring sampling strategies of the queries can be an interesting future direction.
>
> **Q3. “Why not provide YouTube-VIS 2022 experimental results?”**
>
> Thanks for the comment. We focused more on existing datasets because the release date of YouTube-VIS 2022 was very close to the NeurIPS paper submission date. Here, we are glad to share VITA’s accuracy (using a ResNet-50 backbone) on YouTube-VIS 2022 validation set, and we will add more results in revision.
>
> |AP     |AP$_{50}$|AP$_{75}$|AR$_{1}$|AR$_{10}$|
> |:---:  |:---:      |:---:      |:---:      |:---:|
> |39.1   |60.6       |44.3       |35.6       |48.1|

---

### Official Review · Reviewer_4omf · 2022-07-10

**Rating:** 7
**Confidence:** 5
**Soundness:** 3 good
**Presentation:** 3 good
**Contribution:** 3 good

**Summary:**

This paper presents a memory efficient offline video instance segmentation method named VITA. It utilizes object tokens in each frame as condensed frame features which also embed object-level information. On the video level, it collects all the object tokens and use video queries to generate object sequences. The object sequences are multiplied with the frame features to generated object masks similar to MaskFormer work.

**Questions:**

Other questions:
-	Any potential of adapting this method to online VIS?
-	How to determine the number of N_f in the model? Does it have an impact on the performance?


**Limitations:**

Yes.

**Strengths And Weaknesses:**

Strength:
-	The proposed VITA framework achieved state-of-the-art performance on YouTubeVIS and OVIS.
-	Compared to previous transformer-based methods, the proposed method effectively reduces running memory with the image features – object tokens – object sequences pipeline. It is able to run hundreds of frames in a single forward process while existing methods can only accommodate a few dozens.
-	The proposed method has faster convergence compared to existing baseline such as Mask2Former-VIS (Figure 5). Making it an appealing choice for model development.
Weaknesses:
-	Some missing references:
CompFeat: Comprehensive Feature Aggregation for Video Instance Segmentation AAAI 2021

---

> ### Author Response · Authors · 2022-08-02
> **Response to Reviewer 4omf**
>
> We thank the reviewer for recognizing our work “achieves state-of-the-art performance”, “effectively reduces running memory”, “is able to run hundreds of frames”, “has faster convergence”, and “makes an appealing choice for model development”. We appreciate your valuable comments, and we will respond as follows.
>
> ---
>
> **Q1. Missing references.**
>
> Thanks for your kind reminder. We shall cite the paper in revision.
>
> **Q2. Potential to online VIS.**
>
> We appreciate your insightful comment. We believe there is plenty of room for variants of VITA for online inference, and an initial idea could be recurrently updating clip queries using RNNs.
>
> **Q3. Impact of the number of object tokens on the performance.**
>
> N$_{f}$ was set to 100 as the frame-level detector Mask2Former uses 100 queries by default. Please kindly refer to the experimental results in Reviewer-KReH’s section, which analyses the effects of pruning object tokens. Even if we reduce N$_f$ from 100 to 75, the accuracy loss is marginal (-0.1 AP) while the maximum number of frames that VITA can process greatly increases from 1392 to 2635.

---

### Official Review · Reviewer_dieT · 2022-07-12

**Rating:** 6
**Confidence:** 4
**Soundness:** 2 fair
**Presentation:** 3 good
**Contribution:** 1 poor

**Summary:**

The authors applied Mask2former to generate frame-wise object tokens as a pre-processing of a video.  Afterward, the sliding attention mechanism from Swin Transformer is applied to aggregate all frame-wise understanding for  Video Instance Segmentation. The model trains with the losses from DETR and MaskTrackRCNN. The ensemble achieves a state-of-the-art result on Youtube-VIS
and OVIS. This is the first offline method applied to the long video dataset OVIS.

**Questions:**

1. How were the object tokens calculated during the evaluation phase?
2. Is one GPU enough to fit the video into Mask2Former and the proposed method?
3. Why did the method improve performance for the challenging dataset OVIS?
4. What is the FPS of the method compared to prior work?

**Limitations:**

1. The authors claim to have written the limitation in section 5, but it is not the case.

**Strengths And Weaknesses:**

Strengths:
1. The method performs state-of-the-art offline video instance segmentation on two challenging datasets.
2. This is the first time an offline method can run an evaluation on the long video dataset OVIS.
3. The experiments and ablations are sufficient. The writing is good and easy to understand. The method is described well with interpretable figures.

Weaknesses:
1. The primary weakness of this paper is the lack of novelty. It is an ensemble of two prior works, i.e., Mask2Former, and Swin
Transformer. Both of the prior methods have significant performance in their respective objectives. Hence the original contribution of this work is limited.
2. The authors claim to fit long videos in a single GPU, enabling a long-range offline VIS method. But the Mask2Former features are extracted beforehand. Only the object representation from this prior work is processed in a single GPU. So the claim of fitting
long-range video is misleading.
3. There is no qualitative analysis of the model's performance. It is unclear how or why the modules improve performance in some instances of the problem.

---

> ### Author Response · Authors · 2022-08-02
> **Response to Reviewer dieT**
>
> We appreciate your reviews. We are encouraged that you find our work “shows state-of-the-art performance on challenging datasets”, “is the first offline method can run an evaluation on OVIS which consists of long videos”, “demonstrates sufficient experiments and ablations”, “is well written and easy to understand”, and “presents the method with interpretable figures”. Here are our answers to the reviewer’s concerns and questions.
>
> ---
>
> **Q1. Lack of novelty.**
>
> We would like to note that our original contribution is to propose a new paradigm of complete-offline Video Instance Segmentation (i.e., *video-in and video-out*). While existing methods have achieved powerful performance on benchmarks, they lack applications to long and high-resolution videos. Some methods tackle the efficient design, but the inherent property of carrying dense spatio-temporal features leads to limited scalability. Therefore, our motivation is to provide the community with new insights that break the convention. As discussed in ablations (Table. 4 - 6), our proposal brings various choices - inferring long and high-resolution videos, faster convergence when training, and better practicality with a frozen detector.
>
> **Q2. The claim of fitting long-range video.**
>
> Under the complete-offline manner, we first extract the frame-level object queries and mask features by visiting the frame-level image instance segmentation model (Mask2Former [7]). What differs the most from previous works is that VITA can discard heavy intermediate spatio-temporal features. This is because the input of VITA is only a set of frame queries. On the contrary, previous works have to keep dense spatio-temporal features as the features are used for decoding information into object queries. We will add more clarification on this in revision.
>
> **Q3. Qualitative analysis.**
>
> Thank you for the constructive comment. Please kindly refer to Reviewer-56An’s section.
>
> **Q4. Object tokens during the evaluation phase.**
>
> We use entire object tokens decoded by a frame-level object detector during the training and evaluation phases. There is no heuristic to filter it out. As other reviewers (4omf and KReH) also pointed out, it is certainly meaningful to investigate the performance when only a part of object tokens remains. Please kindly refer to the experimental results in Reviewer-KReH’s section, which analyses the effects of pruning object tokens. Even if we reduce N$_{f}$ from 100 to 75, the accuracy loss is marginal (-0.1 AP) while the maximum number of frames that VITA can process greatly increases from 1392 to 2635.
>
> **Q5. Performance improvement on OVIS.**
>
> Thank you for the comment. OVIS (Occluded Video Instance Segmentation), as the name suggests, basically has high occlusion between instances. In addition, as we discussed in section 4.1 in the paper, we observed that OVIS has other three distinct challenges compared to YouTubeVIS. 1) much more instances appear in a single video. 2) the instances with the same categories in the same video have almost similar appearances. 3) much longer video length. While the standard VIS metrics comprehensively reflect these challenges, we suppose that the proposed method achieves high performance on OVIS for the following reason: explicit utilization of object-centric information. We learned a lesson from previous offline methods that building communication between frames is crucial to improving the performance of VIS. The context exchange through memory tokens (e.g., IFC [14]) leads to good performance on YouTubeVIS, which has relatively monotonous motions with few instances. However, IFC may struggle on videos with many instances such as OVIS. We believe that VITA’s object-centric information exchange throughout a video helps to better deal with the challenges in OVIS.
>
> **Q6. FPS comparison.**
>
> Thank you for the question. We measured FPS using V100 GPU with a ResNet-50 backbone. Compared to Mask2Former-VIS [6], which runs at 48.2 FPS, VITA achieves 46.1 FPS. As the VITA module only uses frame queries instead of the whole spatio-temporal features, it shows only marginal speed degradation of 2.1 FPS.
>
> **Limitation.**
>
> We appreciate the feedback, please kindly refer to Reviewer-56An’s section. We shall add the limitation section in the revision.

---

> > ### Comment · Reviewer_dieT · 2022-08-09
> > **Post Rebuttals**
> >
> > I have read the author's response. They try to address all the concerns. I agree with the authors that performance wises it's a new benchmark across all datasets. However, I till think its two-stage processing(1st token extraction,2nd main network) is quite burdensome and stronger features from mask2former help its performance. As it is a really well-written paper with a strong performance but lacks on the earlier points, I will give a weak acceptance.

---

### Official Review · Reviewer_56An · 2022-07-13

**Rating:** 7
**Confidence:** 4
**Soundness:** 3 good
**Presentation:** 4 excellent
**Contribution:** 3 good

**Summary:**

- This paper considers the problem of video instance segmentation through associating the object-centric representation from frame-wise detector.

- The proposed idea uses Transformer (local attentions) to associate the frame-wise object predictions.

- The idea is validated on different VIS benchmarks, e.g. YTB-VIS2019, 2021, and OVIS, showing good results.

**Questions:**

- Numbers in Table2 can be filled in.

-  in Table 6, for frozen detector, what benchmark this is ? VIS-2019 ? please put it in caption.

- if only 20 classes overlap, the recall remains quite high, especially on Swin-L, and this is only AR_10, I imagine AR_100 would be even closer, as you are using 100 queries, so in this case, why the gap on AP is quite large ? for example, 63 vs. 53.4.

- how much VITA can recover the failed prediction from frame-wise predictions ?

**Ethics Review Area:**

["I don’t know"]

**Limitations:**

- There is broader impact, but no limitation section has been provided.

**Strengths And Weaknesses:**

- strengths
    - the paper writing is good, easy to understand.
    - the motivation of building on pre-trained frame wise object detector is good, which enables the model to handle long and high-resolution videos with customer-level GPUs.
    - very good results have been obtained.

- weakness
    - in Table 2, what is the results for Mask2Former on OVIS, they have released model, so I believe it's possible to benchmark these numbers.
    - the analysis can be done more thoroughly, for example, it would be interesting to see, what if the frame-wise object detector fails on some intermediate frames, how much VITA can recover it.
    - not much to say on the weakness.

I generally like this paper, for its reasonable motivation, simple implementation, and good results.

---

> ### Author Response · Authors · 2022-08-02
> **Response to Reviewer 56An**
>
> We thank the reviewer for acknowledging our work and providing helpful comments. We appreciate the remarks that our work “is well written and easy to understand”,  “has good and reasonable motivations”, and “shows good results with a simple implementation”. We will consider the reviewer’s concerns and questions below.
>
> ---
>
> **Q1. Results for Mask2Former on OVIS in Table 2.**
>
> We agree that it is important to compare our method against the state-of-the-art method - Mask2Former. However, we would like to point out that Mask2Former did not officially provide performance for OVIS on paper or in the GitHub repository. We believe this is because it is not possible for Mask2Former to infer the OVIS validation set on top of mainstream GPUs, as the dataset has much longer videos than YouTube-VIS. Therefore, we trained Mask2Former on OVIS by following their official training recipe of YouTube-VIS and we provide the results by running the model using CPU.
>
> |Method|AP|AP$_{50}$|AP$_{75}$|AR$_{1}$|AR$_{10}$|
> |:---|:---:|:---:|:---:|:---:|:---:|
> |Mask2Former-VIS (CPU) [6]|13.6|31.4|10.8|8.8|22.6|
>
> **Q2. Thorough analysis of whether VITA can recover from errors in the frame-level object detector.**
>
> We appreciate the suggestion and agree that analyzing such scenarios would be essential to substantiate the proposed approach.
> * **Qualitative analysis:** Please kindly refer to **Figure. 1 - 2** in **rebuttal_qualitative_results.pdf** in the supplementary materials. We present the predictions of the frame-level object detector and VITA. Each video sample consists of two rows, the top row is the result of the frame-level object detector and the bottom row is the result of VITA. As shown in the results, while the frame-level detector fails to recognize either category or mask of instances that have been largely occluded, our method successfully recovers it by leveraging the temporal information.
> * **Quantitative analysis:** The one way to measure this is to convert the YouTubeVIS validation set into the frame-level annotations, then compare the frame-level AP of the frame-level detector and VITA using COCO API. Unfortunately, there is no publicly available ground truth for the validation set. Although it is not possible to make a concrete comparison, the effect can be demonstrated by restraining exchange of information between different frames in Clip Encoder: setting the window size to *W*=1, AP drops by 1.0 AP.
>
> **Q3. AP gap between end-to-end vs. frozen object detector.**
>
> Compared to COCO, YouTube-VIS dataset is annotated with only a few salient objects in videos as foregrounds. Having weights of the frame-level detector frozen to COCO, the detector cannot adapt to the YouTube-VIS domain and it embeds and interprets more objects in scenes as foregrounds. Therefore, VITA outputs more predictions as a foreground category even if such predictions are not labeled as ground-truths in YouTube-VIS. Please kindly refer to **Figure. 3** in **rebuttal_qualitative_results.pdf** in the supplementary materials. The phenomenon is shown in the following table: the number of outputs that VITA predicts as a foreground category per each threshold.
>
> |Threshold      |0.1    |0.25   |0.5    |0.75|
> |:---           |:---:  |:---:  |:---:  |:---:|
> |Frozen Detector|1116   |855    |674    |530|
> |End-to-End     |653    |579    |542    |504|
>
> As a result, it leads to a lower average precision (AP = ***TP*** / (***TP*** + ***FP***)) as it comes out with more false positive predictions. On the contrary, the more false positive predictions only slightly affect AR by the definition: ***TP*** / (***TP*** + ***FN***).
>
> **Q4. Table 6 Benchmark.**
>
> We appreciate your kind reminder. The benchmark is YouTube-VIS 2019 and we shall add it to the caption in revision.
>
> **Limitation.**
>
> VITA has two major limitations for the ultimate long video understanding. First, the current architecture still has limitations in processing an infinite number of frames. In addition, since object tokens do not explicitly utilize temporal information, they may have difficulties in identifying complex behaviors that span over very long sequences. We believe that devising explicit designs to address these issues will be a promising future direction. We shall add this into the limitation section in the revision.

---

> > ### Comment · Reviewer_56An · 2022-08-09
> > **Decision after rebuttal**
> >
> > Thank you for the reply, it has solved my concerns, I'll keep the score, "Accept".

---

### Meta-Review · Area_Chair_4Bur · 2022-08-26

**Recommendation:** Accept
**Confidence:** Certain

**Metareview:**

All four reviewers are positive about this work (with three Accept and one Weak Accept). All reviewers appreciate the clear writing, solid results, and the idea of using local attentions in transformers to associate object token extracted at each frame. During the discussion phase, the authors further clarified some of the questions and present additional results (e.g., limitations and ablation on frozen/finetuned detector). After reading the reviews and the rebuttal, the AC agrees with the reviewers that this is a solid work with strong results on video instance segmentation. The AC thus recommends to accept.

**Award:**

No

---

### Decision · Program_Chairs · 2022-09-14

Accept